# Local human movement patterns and land use impact exposure to zoonotic malaria in Malaysian Borneo

Kimberly M Fornace[1,2]*, Neal Alexander[3], Tommy R Abidin[4], Paddy M Brock[5], Tock H Chua[4], Indra Vythilingam[6], Heather M Ferguson[5], Benny O Manin[4], Meng L Wong[6], Sui H Ng[4], Jon Cox[1], Chris Drakeley[1]

[1]Faculty of Infectious and Tropical Diseases, London School of Hygiene and Tropical Medicine, London, United Kingdom; [2]Centre on Climate Change and Planetary Health, London School of Hygiene and Tropical Medicine, London, United Kingdom; [3]Department of Infectious Disease Epidemiology, London School of Hygiene and Tropical Medicine, London, United Kingdom; [4]Department of Pathobiology and Medical Diagnostics, Faculty of Medicine and Health Sciences, Universiti Malaysia Sabah, Kota Kinabalu, Malaysia; [5]Institute of Biodiversity, Animal Health and Comparative Medicine, College of Medical, Veterinary and Life Sciences, University of Glasgow, Glasgow, United Kingdom; [6]Parasitology Department, Faculty of Medicine, University of Malaya, Kuala Lumpur, Malaysia

**Abstract** Human movement into insect vector and wildlife reservoir habitats determines zoonotic disease risks; however, few data are available to quantify the impact of land use on pathogen transmission. Here, we utilise GPS tracking devices and novel applications of ecological methods to develop fine-scale models of human space use relative to land cover to assess exposure to the zoonotic malaria *Plasmodium knowlesi* in Malaysian Borneo. Combining data with spatially explicit models of mosquito biting rates, we demonstrate the role of individual heterogeneities in local space use in disease exposure. At a community level, our data indicate that areas close to both secondary forest and houses have the highest probability of human *P. knowlesi* exposure, providing quantitative evidence for the importance of ecotones. Despite higher biting rates in forests, incorporating human movement and space use into exposure estimates illustrates the importance of intensified interactions between pathogens, insect vectors and people around habitat edges.
DOI: https://doi.org/10.7554/eLife.47602.001

*For correspondence:
Kimberly.Fornace@lshtm.ac.uk

Competing interests: The authors declare that no competing interests exist.

## Introduction

Environmental change and human encroachment into wildlife habitats are key drivers in the emergence and transmission of zoonotic diseases (*Lambin et al., 2010*; *Patz et al., 2004*). Individual movements into different habitats influence exposure to disease vectors and animal reservoirs, determining risk and propagation of vector-borne diseases (*Stoddard et al., 2013*; *Stoddard et al., 2009*; *Pindolia et al., 2012*). Increased contact between these populations is theorised to drive increases of the zoonotic malaria *Plasmodium knowlesi* in Malaysian Borneo, now the main cause of human malaria within this region. *P. knowlesi* is carried by long- and pig-tailed macaques (*Macaca fascicularis* and *M. nemestrina*) and transmitted by the *Anopheles leucosphyrus* mosquito group, both populations highly sensitive to land cover and land use change (*Moyes et al., 2016*). Although higher spatial overlap between people, macaques and mosquito vectors likely drives transmission,

the impact of human movement and land use in determining individual infection risks is poorly understood (*Imai et al., 2014*).

The emergence of the zoonotic malaria *Plasmodium knowlesi* has been positively associated with both forest cover and historical deforestation (*Fornace et al., 2016b*; *Shearer et al., 2016*). However, out of necessity, statistical approaches to assess environmental risk factors for *P. knowlesi* and other infectious diseases typically evaluate relationships between disease metrics and local land cover surrounding houses or villages. While an individual may spend most of their time within the vicinity of their residence, this area does not necessarily represent where they are most likely to be exposed to a disease. This is supported by varying associations between *P. knowlesi* occurrence and landscape variables at different distances from households, ranging from 100 m to 5 km, likely partially due to human movement into different surrounding habitats (*Fornace et al., 2016b*; *Brock et al., 2019*). Although land cover variables describing physical terrestrial surfaces are frequently incorporated into disease models, land use is rarely quantified. Land use is commonly defined as 'the arrangements, activities, and inputs that people undertake in certain land cover types' (*IPCC, 2000*). Places with similar types of land cover may be used very differently, with the activities and frequencies with which people visit these places determining the spatial distribution of disease (*Lambin et al., 2010*).

Mathematical modelling studies have revealed the importance of spatial variation in contact rates due to the movement of individuals through heterogeneous environments with varying transmission intensity (*Acevedo et al., 2015*). A multi-species transmission model of *P. knowlesi* highlighted the role of mixing patterns between populations in different ecological settings in determining the basic reproductive rate and subsequent modelling studies illustrate the sensitivity of this disease system to population densities of both people and wildlife hosts (*Imai et al., 2014*; *Yakob et al., 2018*). However, although mechanistic models have been extended to explore the potential importance of these heterogeneities in disease dynamics, there are inherent constraints on model complexity and most models make simplistic assumptions about the habitat uses of different populations.

Empirical data on human population movement is increasingly available, allowing assessment of the impact of mobility on infectious disease dispersion and risks (*Pindolia et al., 2012*). On larger spatial scales, mobile phone data have revealed the role of human migration in the transmission of infectious diseases such as malaria, dengue and rubella (*Chang et al., 2019*; *Wesolowski et al., 2015a*; *Wesolowski et al., 2015b*).Although this data can provide insights into long range movements, spatial resolution of this data is limited, particularly in areas with poor or no mobile coverage, such as forested areas (*Wesolowski et al., 2016*). Alternatively, the advent of low-cost GPS tracking devices allows quantification of fine-scale movements, demonstrating marked heterogeneity in individual movement and risk behaviours (*Stoddard et al., 2013*; *Vazquez-Prokopec et al., 2013*). Combining these data with detailed data on land cover and vector dynamics can provide new insights into how landscapes affect *P. knowlesi* transmission.

Previous studies of *P. knowlesi* have relied on questionnaire surveys, identifying self-reported travel to nearby plantations and forest areas as a risk factor for *P. knowlesi* and other malaria infections (e.g. *Grigg et al., 2017*; *Singh et al., 2004*; *Yasuoka and Levins, 2007*). However, the resultant spatial range and frequency of these movements remain unknown and the definition of different habitat types is entirely subjective. Further, little is known about differences in local movement patterns in different demographic groups. While infections in male adults have been linked to forest and plantation work, it is unknown whether infections reported in women and young children are likely to arise from exposure to similar environments (*Barber et al., 2012*). The main mosquito vector in this area, *An. balabacensis*, is primarily exophagic and has been identified in farm, forest and village areas near houses (*Wong et al., 2015*; *Manin et al., 2016*). Macaque populations are reported in close proximity to human settlements and molecular and modelling studies suggest transmission remains primarily zoonotic in this area (*Imai et al., 2014*; *Lee et al., 2011*; *Chua et al., 2017*). A case control study detected higher abundances of *An. balabacensis* near *P. knowlesi* case households, suggesting the possibility of peri-domestic transmission (*Manin et al., 2016*). Understanding the importance of these habitats is essential to effectively target intervention strategies and predict impacts of future environmental changes.

Key questions remain about where individuals are likely to be exposed to *P. knowlesi* and how landscape determines risk. Functional ecology approaches allow the distribution of different populations to be modelled based on biological resources and relate transmission to landscape and

environmental factors (*Hartemink et al., 2015*). Within wildlife ecology, numerous methods have been developed to estimate utilisation distributions (UDs), the probability of an individual or species being within a specific location during the sampling period (*Papworth et al., 2012*). Although these methods traditionally rely on kernel density smoothing, kernel density estimates may not actually reflect time individuals spend in a specific location if there is substantial missing data or irregular time intervals. Alternatively, biased random bridges (BRBs) improve on these methods by estimating the utilisation distribution as a time-ordered series of points, taking advantage of the autocorrelated nature of GPS tracks to bias movement predictions towards subsequent locations in a time series (*Benhamou, 2011*). This allows for interpolation of missing values and adjustment for spatial error to estimate utilisation distributions representing both the intensity (mean residence time per visit) and frequency of individual visits to specific locations. By integrating these estimates of individual space use with detailed spatial and environmental data in a Bayesian framework, fine-scale patterns of human land use can be predicted and overlaid with spatiotemporal models of mosquito distribution. This allows exploration of how landscape composition, as well as configuration and connectivity between habitats, impacts human exposure to *P. knowlesi* and other vector-borne and zoonotic diseases.

Focusing on one aspect of land use, human movement and time spent within different land cover types, we explored the role of heterogeneity in local space use on disease exposure. Rolling cross-sectional GPS tracking surveys were conducted in two study areas with on-going *P. knowlesi* transmission in Northern Sabah, Malaysia (Matunggong and Limbuak; *Fornace et al., 2018*). We aimed to characterise local movement patterns and identify individuals and locations associated with increased *P. knowlesi* exposure risks by: 1. analysing individual movement patterns and developing predictive maps of human space use relative to spatial and environmental factors, 2. modelling biting rates of the main vector *An. balabacensis*, and 3. assessing exposure risks for *P. knowlesi* based on predicted mosquito and human densities (*Figure 1*) Integrating these three approaches allowed a uniquely spatially explicit examination of disease risk.

## Materials and methods

### Study site

This study was conducted in two rural communities in Northern Sabah, Malaysia: Matunggong, Kudat (6˚47N, 116˚48E, population: 1260) and Limbuak, Pulau Banggi (7˚09N, 117˚05E, population: 1009) (*Figure 2*). These areas were the focus for integrated entomology, primatology and social science studies for risk factors for *P. knowlesi* (https://www.lshtm.ac.uk/research/centres-projects-groups/monkeybar), with clinical cases and submicroscopic infections reported from both sites and *P. knowlesi* sero-prevalence estimated as 6.8% and 11.7% in Matunggong and Limbuak, respectively (*Fornace et al., 2018*).

Demographic data and GPS locations of primary residences were collected for all individuals residing in these areas (*Fornace et al., 2018*). Potential spatial and environmental covariates for these sites were assembled from ground-based and remote-sensing data sources (*Supplementary file 1*). The enhanced vegetation index (EVI) was used to capture temporal changes in vegetation levels; this index captures photosynthetic activity and has higher sensitivity in high biomass areas compared to the normalised difference vegetation index (NDVI) frequently used. Due to the high cloud cover within this area, EVI at a high spatial resolution could not be obtained for all time periods. Instead, EVI data at a lower spatial but higher temporal resolution was used and monthly averages were calculated from all available cloud-free data and resampled to 30 m per pixel (*Didan, 2015*).

### GPS tracking survey

A minimum of 50 participants per site were targeted in a rolling cross-sectional survey (*Johnston and Brady, 2002*). During pre-defined two-week intervals, randomly selected participants from comprehensive lists of eligible community members were asked to carry a QStarz BT-QT13000XT GPS tracking device (QStarz, Taipei, Taiwan) programmed to record coordinates continuously at one-minute intervals for at least 14 days regardless of individual movement. Individuals were excluded if they were not primarily residing in the study area, under 8 years old or did not

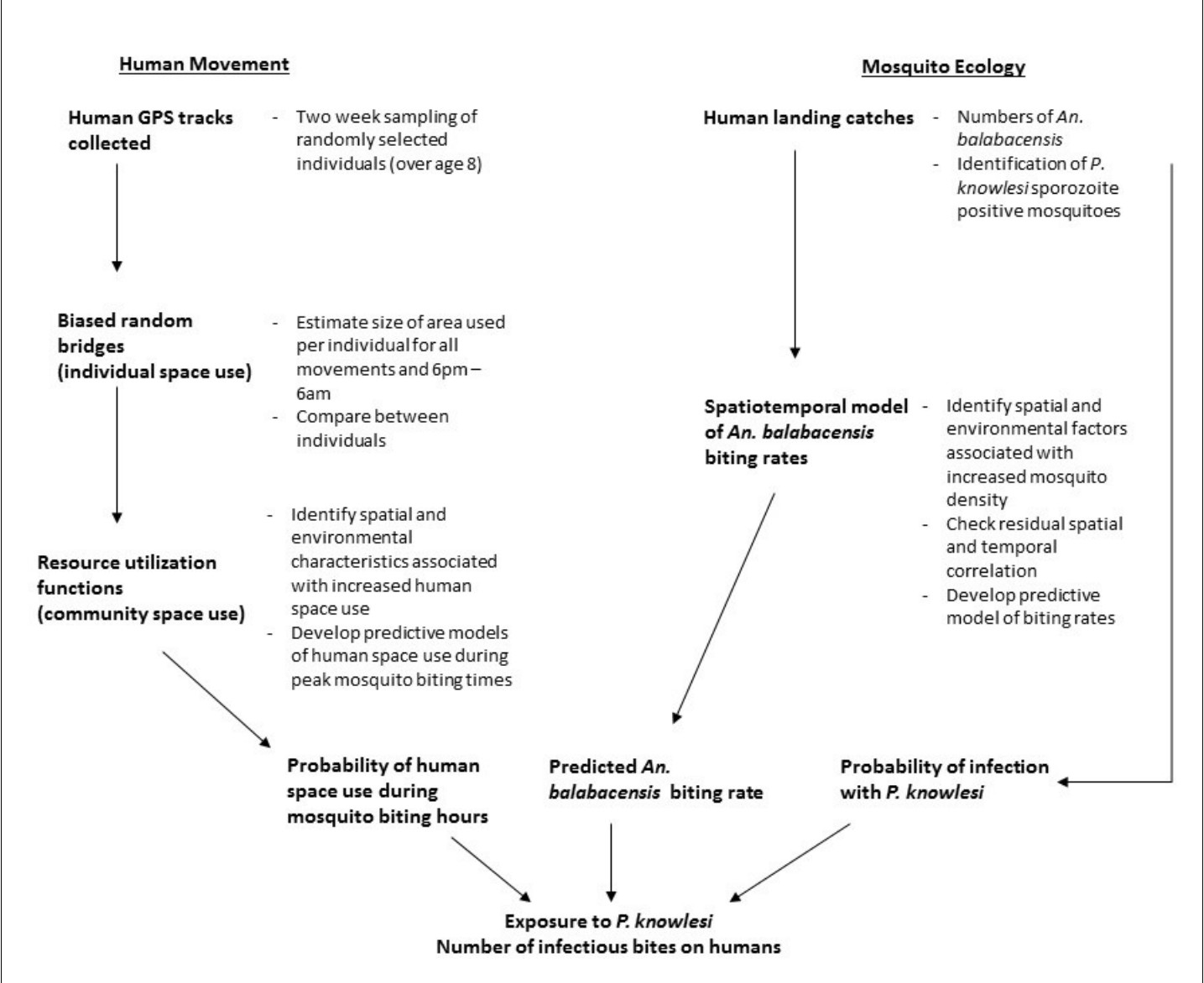

**Figure 1.** Analysis methods used to estimate individual and community-level exposure to *P. knowlesi* sporozoite positive *An. balabacencis* bites.
DOI: https://doi.org/10.7554/eLife.47602.002

consent. Trained fieldworkers visited the participant every two days to confirm the device was functioning, replace batteries and administer questionnaires on locations visited and GPS use. Fieldworkers recorded whether the device was working and if the individual was observed carrying the GPS device to assess compliance. Individuals were excluded from analysis if insufficient GPS data were collected (less than 33% of sampling period) or individuals were observed not using the device for two or more visits.

## Human space use

Biased random bridges were used to calculate individual utilisation distributions, the probability of an individual being in a location in space within the sampled time period (*Benhamou, 2011*). Within this study, large proportions of GPS fixes were missed due to technical issues with batteries and GPS tracking; biased random bridges were used to interpolate between known locations and adjust for missing data, using the time series GPS data to provide a more accurate estimate of space use.

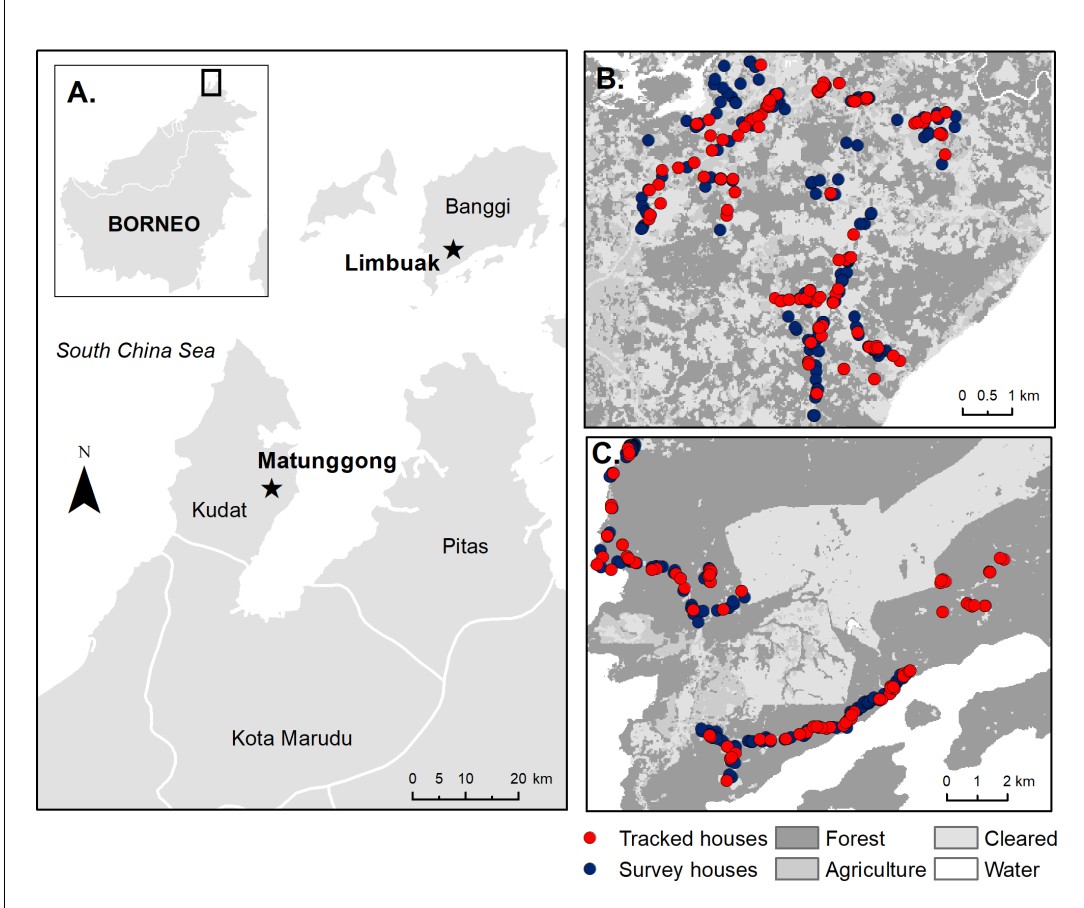

**Figure 2.** Study sites and sampled houses. (A) Location of study sites and tracked houses (households with one or more individual GPS tracked) and survey houses (households with only questionnaire data collected and used for prediction) in (B) Matunggong, Kudat and (C) Limbuak, Banggi; description of land cover classification and survey methodology in *Fornace et al. (2018)*.

DOI: https://doi.org/10.7554/eLife.47602.003

Utilisation distributions were calculated separately for each individual for all movement and night-time only movements (6pm – 6am).

To fit biased random bridges, we estimated the maximum threshold between points before they were considered uncorrelated ($T_{max}$) as 3 hr based on typical reported activity times. The minimum distance between relocations ($L_{min}$), the distance below which an individual is considered stationary, was set at 10 m to account for GPS recording error based on static tests. Finally, the minimum smoothing parameter ($h_{min}$), the minimum standard deviation in relocation uncertainty, was set as 30 m to account for the resolution of habitat data and capture the range of locations an individual could occupy while being recorded at the same place (*Papworth et al., 2012*; *Benhamou, 2011*). Estimates of the core utilisation area (home range) were based on the 99th percentile, representing the area with a 99% cumulative probability distribution of use by the sampled individual.

To assess relationships between space use and environmental factors and develop predictive maps of community space use, we fit resource utilisation functions, regression models in which the utilisation distributions are used as the response variable, improving on models using raw GPS count points as the response when there is location uncertainty and missing data (*Hooten et al., 2013*). The probability density function (utilisation distribution) per individual was rasterised to 30 m$^2$ grid cells and environmental and spatial covariates extracted for each grid cell. Potential environmental covariates included distance to the individual's own house, distance to closest house, distance to roads, land use class (forest, agriculture, cleared or water), distance to forest edge, elevation and slope (*Supplementary file 1*). Resource utilisation was modelled as a Bayesian semi-continuous

(hurdle) model with two functionally independent components, a Bernoulli distribution for the probability of individual $i$ visiting a specific grid cell $j$ ($\omega_{ij}$) and a gamma distribution for the utilisation distribution in grid cells visited ($y_{ij}$) (*Blangiardo and Spatial, 2015*; *Sadykova et al., 2017*). For each individual, we defined absences to be all grid cells with a utilisation distribution less than 0.00001, indicating a very low probability the individual visited this grid cell during the study period. We included all presences (grid cells with a utilisation distribution >0.00001) and randomly subsampled equal numbers of absences (grid cells not visited) for each individual as including equal numbers of presences and absences can improve predictive abilities of species distribution models (*Barbet-Massin et al., 2012*). The utilisation distribution for grid cells visited is defined as:

$$y_{ij} = \left\{ \begin{array}{l} \mathrm{Gamma}\left(\frac{\mu_{ij}^2}{\sigma^2}, \frac{\sigma^2}{\mu_{ij}}\right) \text{with probability } 1 - \phi_{ij} \\ 0 \text{ with probability } \phi_{ij} \end{array} \right\}$$

Where the mean of $y_{ij}$ is given by:

$$\mu_{ij} = \mathrm{E}\left(y_{ij}|X_{ij}^T\right) = \left(1 - \omega_{ij}\right)\upsilon_{ij}$$

The full model was specified as:

$$\omega_{ij} \sim \mathrm{Bernoulli}\left(\phi_{ij}\right)$$

With the linear predictor for the Bernoulli model specified as:

$$\mathrm{logit}\left(\phi_{ij}\right) = \beta_0 + X_{ij}^T\beta_i + \gamma_j$$

Where $\beta_0$ represents the intercept, $X_{ij}^T\beta_i$ represents a vector of covariate effects and $\gamma_j$ represents the additive terms of random effects for individual. For the Gamma component, $\sigma^2$ is the variance and the linear predictor $\upsilon_{ij}$ is specified as:

$$\log\left(\upsilon_{ij}\right) = \alpha_0 + X_{ij}^T\alpha_i + \varphi_j$$

With $\alpha_0$ representing the intercept, $X_{ij}^T\alpha_i$ representing a vector of coordinates and $\varphi_j$ representing the random effects. Weakly informative normal priors specified as Normal (0,1/0.01) were used for all intercepts and coefficients. Bayesian inference was implemented using integrated nested Laplace approximation (INLA) (*Rue et al., 2009*). This approach uses a deterministic algorithm for Bayesian inference, increasing computational efficiency relative to Markov chain Monte Carlo and other simulation-based approaches (*Blangiardo and Spatial, 2015*). We did not explicitly include spatial autocorrelation as several distance-based covariates were included (e.g. distance from own house) (*Hooten et al., 2013*). Predictive models used data for all individuals aged 8 or over residing in these communities (*Table 1*) and models were limited to land areas within 5km of households included in the study site. Separate models were fit for each site.

## Exposure to infected vectors

To estimate vector biting rates, we assembled data from 328 nights of human landing catches (HLCs) conducted with 5 km of the Matunggong study site while GPS tracking was on-going, including: monthly longitudinal surveillance (*Wong et al., 2015*), investigations surrounding households of cases and controls (*Manin et al., 2016*), and environmentally stratified outdoor catches (*Sh et al., 2016*) (*Supplementary file 2*). We limited this data to counts of *An. balabacensis*, the primary *knowlesi* vector, which comprises over 95% of *Anopheles* caught in this region. As one experiment only collected mosquitoes for 6 hr, we fit a linear model of all available data vs totals after 6 hr catches to estimate the total numbers of *An. balabacensis* which would have been caught over 12 hr for these data ($R^2$ = 0.85). Plausible environmental covariates were assembled, including land use type, slope, aspect, elevation, topographic wetness index, EVI, population density and average monthly temperature and rainfall. To select variables for inclusion, Pearson correlation analysis was used to assess multicollinearity between selected environmental variables. As topographic slope and TWI had a strong negative correlation, only TWI was included in the analysis. The autocorrelation function

**Table 1.** Baseline characteristics of study site communities and sampled populations

|  | Matunggong |  | Limbuak |  |
|---|---|---|---|---|
|  | Sampled | Community* | Sampled | Community* |
| N | 134 | 958 | 109 | 633 |
| Gender |  |  |  |  |
| Male, % (n) | 51.5% (69) | 46.1% (442) | 47.7% (52) | 46.1% (292) |
| Women, % (n) | 48.5% (65) | 53.9% (516) | 52.3% (57) | 53.9% (341) |
| Age in years, median (IQR) | 31 (17–53) | 32.5 (8–51) | 29 (15–46) | 30 (15–47) |
| Main occupation, % (n) |  |  |  |  |
| Farming | 29.9% (40) | 28.6% (274) | 7.3% (8) | 10.2% (65) |
| Plantation work | 10.4% (14) | 8.6% (82) | 10.1% (11) | 7.6% (48) |
| Student | 26.1% (35) | 27.7% (265) | 26.6% (29) | 21.0% (133) |
| Other | 6.7% (9) | 9.1% (87) | 15.6% (17) | 14.4% (91) |
| No employment/housewife | 26.9% (36) | 26.1% (250) | 40.4% (44) | 46.8% (296) |

*Community includes all individuals eligible for these surveys (residents ages eight and over).

DOI: https://doi.org/10.7554/eLife.47602.004

(ACF) and partial autocorrelation function (PACF) were used to explore correlation between time lags.

A Bayesian hierarchical spatiotemporal model was implemented using counts of *An. balabacensis* bites as the outcome, denoted as $m_{it}$; $j = 1\ldots n$; $t = 1\ldots n$; where $j$ indexes location and $t$ indexes month. The log number of person-nights per catch was included as an offset to adjust for numbers of catchers conducting HLCs during different experiments. As the data were overdispersed, a negative binomial distribution was used to model $m_{it}$. The linear predictor was specified as:

$$\log(\mu_{jt}) = \log(N_{jt}) + Z_0 + D_{jt}^T Z + w_j + e_t$$

Where $N_{ijt}$ represents the number of person-nights for each HLC catch, $Z_0$ represents the intercept, $D_{jt}^T Z$ represents a vector of covariates, $w_j$ is the spatial effect and $e_t$ is the temporal effect. The temporal effect $e_t$ was included as a fixed effect, random effect or temporally structured random walk model of order 1 in candidate models (*Lindgren and Rue, 2015*). The spatial effect $w_j$ was modelled as a Matern covariance function between locations $s_j$ and $s_k$:

$$W \sim \textit{Multivariate Normal } (0, \Sigma)$$

$$\Sigma_{hk} = Cov(\xi(s_h), \xi(s_k)) = Cov(\xi_h, \xi_k) = \frac{\sigma^2}{\Gamma(\lambda)2^{\lambda-1}} (\kappa||s_h - s_k||)^\lambda K_\lambda(\kappa||s_h - s_k||)$$

Where $||s_h - s_k||$ denotes the Euclidean distance between locations $s_h$ and $s_k$, $\xi(s)$ is the latent Gaussian field accounting for spatial correlation, $\sigma^2$ is the spatial process variance and $K_\lambda$ is a modified Bessel function of the second kind and order $\lambda > 0$. $\kappa$ is a scaling parameter related to $r$, the distance at which spatial correlation becomes negligible, by $r = \sqrt{8\lambda}/\kappa$. A stochastic partial differential equations (SPDE) approach was used, representing the spatial process by Gaussian Markov random fields (GMRF) by partitioning the study area into non-intersecting triangles (*Lindgren et al., 2011*). This approach represents the covariance matrix $\Sigma$ by the inverse of the precision matrix $Q$ of the GMRF (*Blangiardo and Spatial, 2015*; *Lindgren et al., 2011*). Prior distributions were specified on fixed effects and hyperparameters. A vague normal prior distribution was used for the intercept. Weakly informative priors were used for fixed effects specified as *N(1,1/0.01)*. Priors for spatial hyperparameters were specified as range $r \sim N(10, 1/0.01)$ and standard deviation $\sigma \sim N(0.1, 1/0.01)$ as described by Lindgren and Rue (*Lindgren and Rue, 2015*).

As these vectors are rarely reported indoors (*Manin et al., 2016*) and HLCs were primarily conducted outside, we excluded areas within houses for calculations of exposure risks. The proportion of infectious mosquitoes, $c$, was parameterised using a beta distribution for *P. knowlesi* sporozoite

rates within this site; with only 4 out of 1524 collected mosquitoes positive, it was not possible to look at variations of infection rates by time and space. Spatially explicit exposure risks were calculated as derived quantity from human resource utilisation, mosquito biting rate models and probability of *P. knowlesi* sporozoite positivity. Individual exposure risk was explored using a simple exposure assessment model where the number of infected bites received by an individual is the sum of bites by infected vector across all locations visited, with the number of infectious bites received by individual *i* in month *t* as:

$$r_{it} = c \sum_{j=1}^{J} y_{ij} m_{jt}$$

Where *j* indexes the grid cells visited, $y_{ij}$ is the utilisation distribution, $m_{jt}$ is the number of bites per individual in that cell and month, and *c* is the proportion of infectious mosquitoes (**Stoddard et al., 2009**). To evaluate places associated with exposure for the entire community, we calculated the number of infectious bites per grid cell each month as:

$$r_{jt} = c \sum_{i=1}^{I} Y_{ij} m_{jt}$$

Where $Y_{ij}$ is the predicted utilisation distribution for all individuals within the community per grid cell *j*. All analyses were conducted in R version 3.5, with Bayesian models implemented using Integrated Nested Laplace Approximation (INLA) (**Rue et al., 2009**). Model fit was assessed using deviance information criteria (DIC) and area under the receiver operating curve (AUC), root mean square error (RMSE) or conditional predictive ordinate (CPO) (**Held et al., 2010**).

## Ethics approval

This study was approved by the Medical Research Sub-Committee of the Malaysian Ministry of Health (NMRR-12-537-12568) and the Research Ethics Committee of the London School of Hygiene and Tropical Medicine (6531). Written informed consent was obtained from all participants or parents or guardians and assent obtained from children under 18.

## Results

Between February 2014 and May 2016, 285 consenting people participated in the GPS tracking study with 243 included in the final analysis including 109 in Limbuak and 134 in Matunggong (**Table 1**). The most commonly reported occupation was farm or plantation work (n = 73), primarily conducted within the immediate vicinity of the house. A total of 3,424,913 GPS points were collected, representing 6,319,885 person-minutes of sampling time. Median sampling duration was 16.27 days (IQR 13.72–19.97), with points recorded for a median of 59.1% (IQR: 46.9–71.1%) of the sampling duration. Maximum distances travelled ranged from no travel outside the house to 116 km, with a median distance travelled of 1.8 km. Utilisation distributions (UDs), the probability of an individual being in a location in space within a given time (**Figure 3**), varied by gender and occupation. Individuals at the more rural Limbuak site covered larger distances (**Table 2**), with the largest distances covered by individuals reporting primary occupations of fishing (n = 5) and office work (n = 9). Although substantial differences were reported in all movements (24 hr sampling) between seasons, no seasonal differences were observed in human movements during peak *Anopheles* biting times (6pm-6am).

For both study areas, we developed models of community space use during peak mosquito biting hours (6pm – 6am), in the form of resource utilisation functions, predictions of time- and space-specific UDs on the basis of spatial and environmental variables (**Papworth et al., 2012**). Between 6pm – 6am, human space use (UDs) was mostly predictable and negatively correlated with distance from the individual's house, other houses, roads and slope. The AUC for presence/absence models was 0.936 for Matunggong and 0.938 for Limbuak and RMSE for the overall model was 0.0073 and 0.0043 for Matunggong and Limbuak, respectively. While individuals were more likely to use areas further away from forests in the Matunggong site, human space use was positively correlated with proximity to forests in the Limbuak site (**Table 3**). Despite marked differences between different

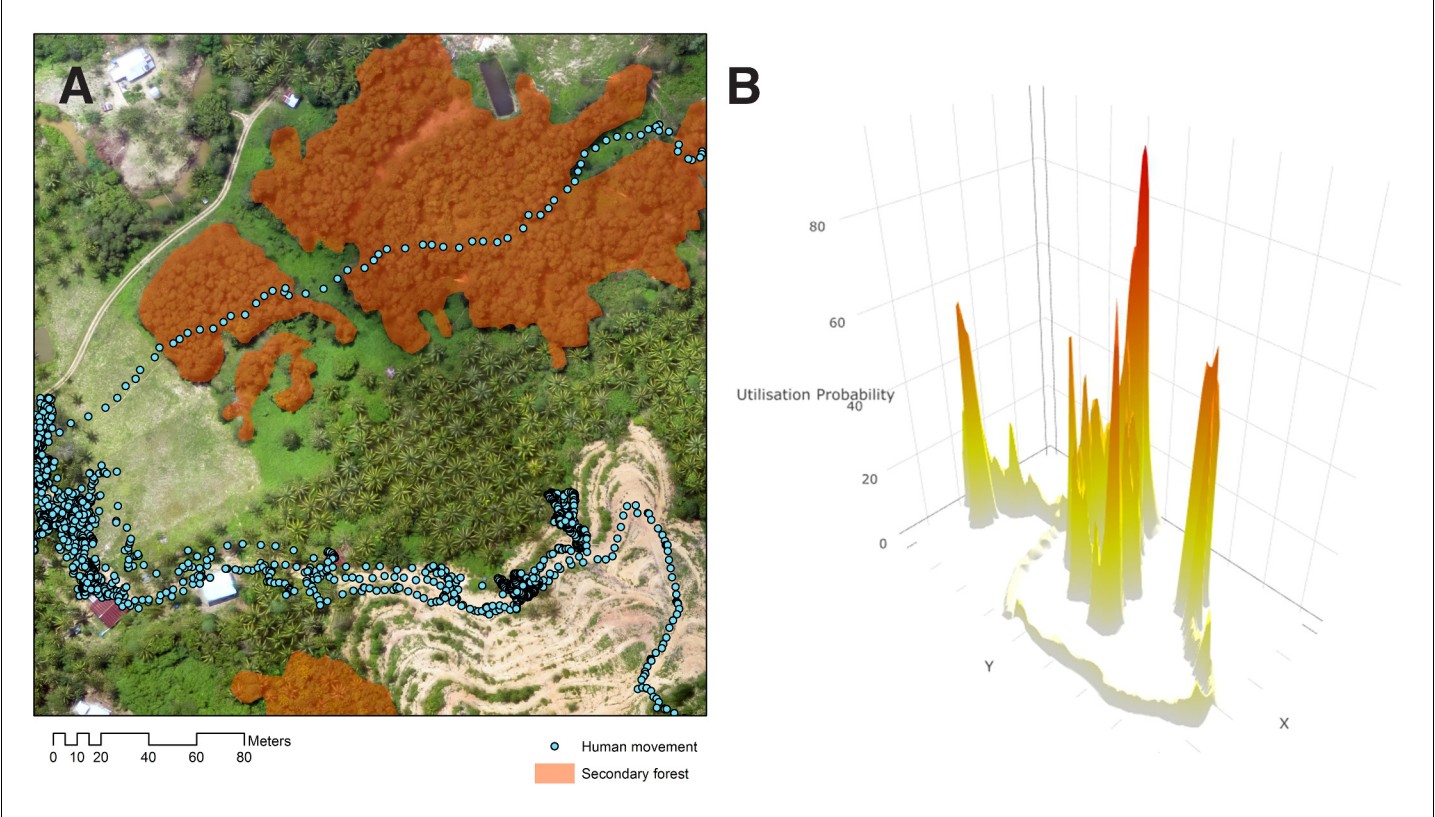

**Figure 3.** Human movement relative to habitat. (A) Example of GPS tracks from a 22-year-old male plantation worker in Matunggong over aerial imagery, (B) Probability density of an individual utilisation distribution calculated from GPS tracks.
DOI: https://doi.org/10.7554/eLife.47602.005

demographic groups and seasons observed during 24 hr movements, these factors did not improve the predictive power of the model for movements between 6pm and 6am.

Between August 2013 and December 2015, 4814 *An. balabacensis* were caught from 328 sampling nights in 155 unique locations. The median biting rate was 2.1 bites per night per person, ranging from 0 to 28 bites per person per night (*Figure 4*). Despite monthly variation, including temporal autocorrelation did not improve model fit (*Table 4*). Although no associations were identified between land classification and vector density in this site, models identified positive relationships with enhanced vegetation indices (EVI) and negative associations with distance to forest and human population density (*Table 5*). Of 1524 mosquitoes tested for *Plasmodium* sporozoites, the median sporozoite rate was 0.24% (95% CI: 0.09–0.58%).

For individuals included in the GPS tracking study in Matunggong, where both human movement and entomology data were available, we calculated exposure risks as a derived quantity from utilisation distributions and mosquito biting rate models. Exposure varied markedly between individuals, with an overall 150-fold difference in predicted mean probabilities of infected bites per night (range: 0.00005–0.0078) (*Table 6*). No clear differences were observed between genders, age groups or occupations of individuals sampled and there was no association between risk and distance travelled.

Using the resource utilisation function with demographic and spatial data for all individuals in Matunggong, we predicted community-wide space use and estimated exposure to infected mosquitoes (*Figure 5*). The predicted number of person nights per grid cell for the entire community ranged from 0 to 12.79 (median: 0.01, IQR: 0.0004–0.99), with the mean probability of a community member exposed to an infected bite per grid cell of 0.00082 (IQR: 0.00001, 0.00050). Although over 43% of the study site is forest and relatively high biting rates were predicted in forests during the study period (mean: 1.94, range: 0.04–12.59), this habitat was rarely used by people in the evenings,

**Table 2.** Home range estimates by demographic group and site

| | Area of 99% UD for all movement (hectares) Median (IQR) | Area of 99% UD from 6pm – 6am (hectares) Median (IQR) |
|---|---|---|
| **Demographic group** | | |
| Men | 32.09 (7.07, 148.93) | 4.50 (2.79, 19.53) |
| Women | 74.25 (12.24, 320.74) | 6.08 (2.79, 24.17) |
| Children (under 15) | 26.01 (6.39, 151.94) | 3.83 (2.79, 8.73) |
| **Occupation** | | |
| Farming | 29.34 (8.15, 324.38) | 6.75 (2.79, 19.80) |
| Plantation work | 49.14 (9.72, 201.33) | 4.59 (2.79, 27.72) |
| Fishing | 442.49 (40.07, 1189.00) | 227.16 (4.05, 465.14) |
| Office work | 96.80 (63.61, 256.75) | 13.63 (2.88, 20.14) |
| Other | 19.98 (6.30, 26.82) | 2.97 (2.61, 18.27) |
| No employment/housewife | 43.38 (11.97, 157.59) | 3.60 (2.79, 19.12) |
| **Site** | | |
| Limbuak | 99.99 (24.57, 387.54) | 7.74 (2.88, 58.05) |
| Matunggong | 12.02 (3.94, 85.55) | 2.97 (2.70, 11.77) |
| **Season** | | |
| Dry (February – July) | 28.62 (5.45, 252.45) | 4.19 (2.79, 19.60) |
| Wet (August – January) | 54.90 (17.23, 160.99) | 4.64 (2.79, 19.35) |

DOI: https://doi.org/10.7554/eLife.47602.006

with less than 8% of predicted person-nights in forests. Models only based on mosquito biting rates and not including human space use predicted 42% of infectious bites occurred in forested areas and only 8.6% of bites occurring within 100 m of houses (*Figure 5C*). In contrast, when space use patterns are included, over 91% of predicted infected bites were predicted within 500 m of houses

**Table 3.** Estimated coefficients for fixed effects of resource utilisation functions (6pm – 6am).

| | Matunggong | | | Limbuak | | |
|---|---|---|---|---|---|---|
| | Mean | SD | 95% CI | Mean | SD | 95% CI |
| **Probability of presence/absence** | | | | | | |
| Intercept | 3.383 | 0.839 | 3.218, 3.547 | 3.571 | 0.104 | 3.368, 3.775 |
| Distance from own house (km) | −0.954 | 0.006 | −0.966,−0.942 | −0.543 | 0.003 | −0.548,−0.539 |
| Distance from forest (km) | 5.997 | 0.177 | −5.650, 6.344 | −1.845 | 0.050 | −1.944,−1.746 |
| Distance from road (km) | −5.552 | 0.057 | −5.663,−5.441 | −3.656 | 0.019 | −3.694,−3.618 |
| Distance from houses (km) | −0.504 | 0.030 | −0.563,−0.444 | 0.176 | 0.007 | 0.162, 0.189 |
| Elevation (100 MSL) | −0.710 | 0.025 | −0.759,−0.662 | −1.268 | 0.037 | −1.340,−1.197 |
| Slope (degrees) | −0.0244 | 0.002 | −0.028,−0.021 | −0.009 | 0.001 | −0.012,−0.006 |
| **Utilisation distributions for locations present** | | | | | | |
| Intercept | −6.846 | 0.866 | −8.549,−5.147 | −5.676 | 1.017 | −7.673,−3.681 |
| Distance from own house (km) | −0.583 | 0.004 | −0.590,−0.576 | −0.308 | 0.002 | −0.311,−0.305 |
| Distance from forest (km) | 12.012 | 0.199 | 11.621, 12.403 | −1.771 | 0.049 | −1.868,−1.675 |
| Distance from road (km) | −0.833 | 0.054 | −0.939,−0.728 | −1.532 | 0.011 | −1.554,−1.511 |
| Distance from houses (km) | −0.819 | 0.023 | −0.864,−0.773 | −0.239 | 0.006 | −0.249,−0.228 |
| Elevation (100 MSL) | 0.664 | 0.027 | 0.610, 0.718 | −0.297 | 0.003 | −0.303,−0.297 |
| Slope (degrees) | −0.021 | 0.002 | −0.024,−0.018 | −0.034 | 0.001 | −0.036,−0.031 |

DOI: https://doi.org/10.7554/eLife.47602.007

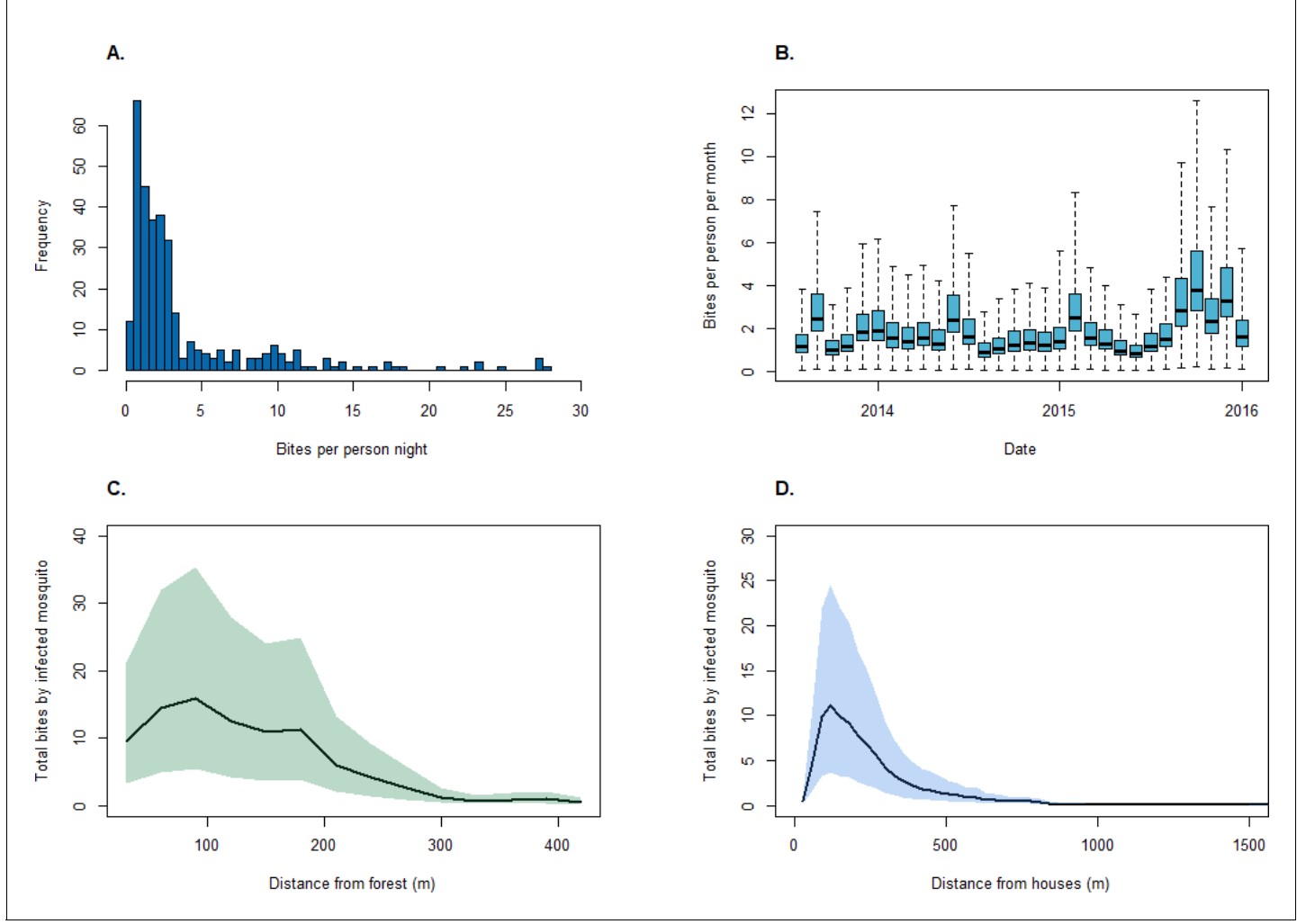

**Figure 4.** Mosquito biting rates. (A) *An. balabacensis* biting rate per person-night from data collected in Matunggong, (B) Predicted mean *An. balabacensis* biting rates per month from spatiotemporal models, (C) Predicted number of bites for all individuals residing in Matunggong by distance from secondary forest, and by (D) Distance from households.

DOI: https://doi.org/10.7554/eLife.47602.009

(*Figure 5D*). Highest exposure risks were consistently found near forest edges and in close proximity to households, despite spatial and temporal heterogeneity and model uncertainty (*Figure 4*).

**Table 4.** Model selection statistics for mosquito biting rates

| Model | | DIC* | Marginal likelihood | Model complexity* | RMSE* | Mean log-score (CPO) |
|---|---|---|---|---|---|---|
| M1 | No spatial or temporal effect | 2367.03 | −1196.61 | 4.12 | 4.99 | 3.61 |
| M2 | Spatial effect only | 2292.97 | −1175.47 | 40.03 | 4.42 | 4.16 |
| M3 | Spatial effect + month as fixed effect | 2282.88 | −1173.68 | 43.99 | 4.24 | 3.90 |
| M4 | Spatial effect + month as random effect | 2222.89 | −1155.91 | 50.28 | 4.05 | 3.61 |
| M5 | Spatial effect + month as random walk | 2225.43 | −1167.79 | 47.55 | 4.09 | 3.63 |

DOI: https://doi.org/10.7554/eLife.47602.008

**Table 5.** Posterior rate ratio estimates and 95% Bayesian credible interval (BCI) for model 4 of mosquito biting rates.

| Covariate | Mean | 95% BCI Rate Ratio 2.5% | 97.5% |
|---|---|---|---|
| Population density | 0.963 | 0.916 | 1.004 |
| EVI | 3.185 | 1.185 | 8.532 |
| Distance to forest (100 m) | 0.926 | 0.871 | 0.976 |
| Spatial range (km) | 3.120 | 0.514 | 6.926 |

DOI: https://doi.org/10.7554/eLife.47602.010

## Discussion

This study highlights the importance of human space use in different land cover types in determining exposure to zoonotic and vector-borne diseases such as *P. knowlesi*. Although *P. knowlesi* has previously been associated with forest exposure (e.g. *Grigg et al., 2017*) and higher biting rates have been reported in forest interiors (*Wong et al., 2015*), this novel approach incorporating both mosquito and human space use data provides a new perspective on peri-domestic transmission, with more than 90% of infectious bites predicted in areas surrounding households at forest edges. This study additionally demonstrates the utility of ecological methods to understand human movement and identify geographical areas associated with higher contact with disease vectors.

Within these communities, local movement patterns during peak vector times were largely predictable and could be explained by spatial and environmental factors. However, despite this finding, there was substantial variation in predicted exposure between individuals as a result of heterogeneity in habitats used. No significant differences in exposure were predicted between men and women, with individuals with high exposure risks identified across occupational and age groups. Although this finding differs from clinical reports, a comprehensive survey within this community identified equal proportions of men and women exposed to *P. knowlesi* as evidenced by specific antibody responses and data on asymptomatic infections suggests higher numbers of non-clinical infections in women (*Fornace et al., 2018*; *Fornace et al., 2016a*). While infrequent events or long-range movements (such as hunting trips) may contribute to these differences in clinical cases and may not have been captured within this two-week sampling period within the study site, this analysis highlights the importance of routine movements into local environments in shaping exposure risks.

This improved understanding of how local human land use is related to exposure risk has important implications for surveillance and control programmes. Malaria control programmes often rely on interventions within the house, such as insecticide treated bednets and indoor residual spraying; however, movements outside during peak biting times illustrate the importance of also targeting outdoor transmission. The identification of areas where exposure is likely to occur can further be

**Table 6.** Probabilities of infected bites per person per night for sampled individuals in Matunggong by demographic characteristics.

| | Predicted infectious bites per night (median [IQR]) |
|---|---|
| **Demographic group** | |
| Men | 0.00157 (0.000804, 0.00289) |
| Women | 0.00219 (0.000864, 0.00307) |
| Children (under 15) | 0.00131 (0.000812, 0.00330) |
| **Occupation** | |
| Farming | 0.00180 (0.00101, 0.00362) |
| Plantation work | 0.00216 (0.000680, 0.00278) |
| Student | 0.00143 (0.000915, 0.00304) |
| Other | 0.00225 (0.000852, 0.00302) |
| No employment/housewife | 0.00142 (0.000297, 0.00263) |

DOI: https://doi.org/10.7554/eLife.47602.011

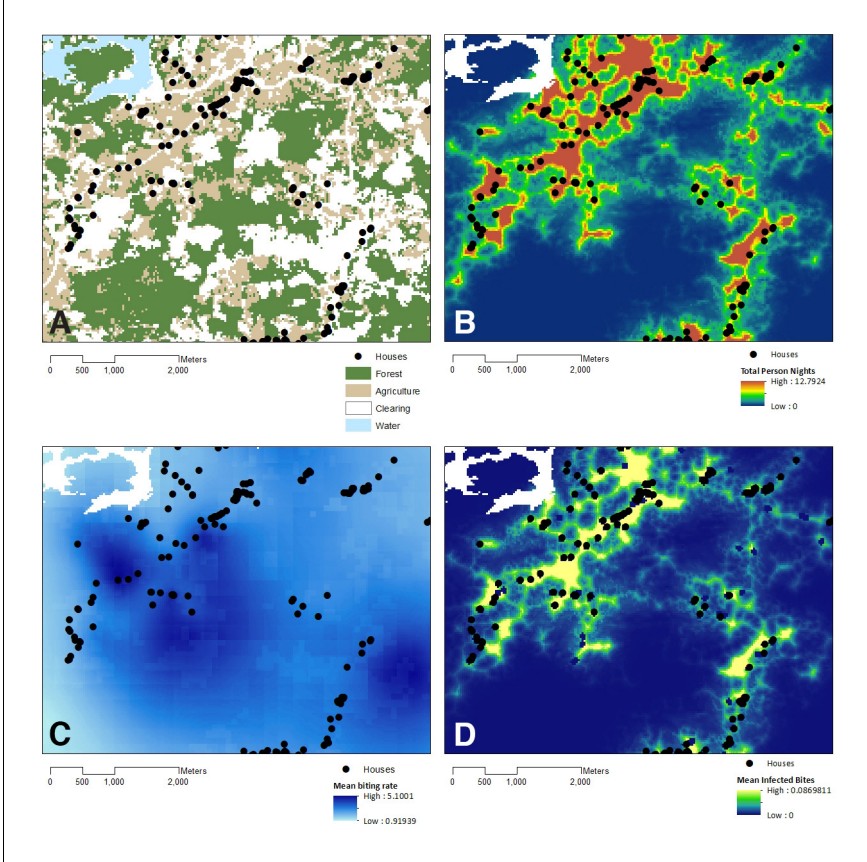

**Figure 5.** Model outputs relative to land cover. (**A**) Land use in Matunggong site, (**B**) Predicted number of person-nights for entire community per grid cell, (**C**) Predicted mosquito biting rates, (**D**) Predicted infected bites per grid cell.

DOI: https://doi.org/10.7554/eLife.47602.012

used to refine interventions; for example, although insecticide treated hammocks have been proposed for deep forest environments, larval source management may be more appropriate to target environments in close proximity to houses. Although initial *P. knowlesi* cases were primarily identified in adult men living and working in forests (*Singh et al., 2004*), this study illustrates the potential importance of peri-domestic habitats in transmission and provides quantitative insight on mixing between people and infected mosquitoes in forest fringe areas. As Malaysia moves towards malaria elimination, surveillance systems are incorporating novel focal investigation methods, including monitoring changes in local land use and populations at risk (*Bahagian Kawalan Penyakit, Kementerian Kesihatan Malaysia, 2016*). In additional to routine vector surveillance, this study highlights the need to incorporate measures of human space when defining risk zones.

Even with the large and highly detailed movement dataset analysed, this study was limited by the availability of mosquito data; as human landing catch data were assembled from other studies, there was not uniform spatial and temporal coverage of the study site increasing uncertainty. The limited mosquito data availability precluded development of mosquito biting rate models for Limbuak and other outlying islands. An additional limitation to estimating mosquito biting rates was the difficulty obtaining spatially and temporally resolute remote sensing data for predictors due to high cloud cover (*Weiss et al., 2015*). As few positive mosquitoes were identified, uniform estimates of sporozoite rates based on available data were used across the Matunggong site; if further data were available, these models could be refined to incorporate estimates of human and macaque density, mosquito biting preferences in different habitats and infection levels in all hosts (*Yakob et al., 2010*). Additionally, as this study was designed to quantitatively estimate time spent in different

landscapes, further studies could explore other aspects of land use, such as the purposes of travel, activities undertaken or practices used to modify or management land cover.

Despite these limitations, this is the first large-scale study to utilise GPS tracking data and ecological methods to create fine-scale maps of exposure risk. This study highlights the importance of incorporating heterogenous patterns of human space use into disease models, as the majority of human exposure may occur in areas with lower vector biting rates but greater probabilities of human use. Further, results quantitatively illustrate the importance of forest edges and local habitat in *P. knowlesi* transmission and can inform understanding of other zoonotic and vector-borne diseases.

## Acknowledgements

We would like to thank the MONKEYBAR project team and the participants in Sabah, Malaysia for their help with these studies. We acknowledge the Medical Research Council, Natural Environmental Research Council, Economic and Social Research Council and Biotechnology and Biosciences Research Council for funding received for this project through the Environmental and Social Ecology of Human Infectious Diseases Initiative, grant no. G1100796. This work was additionally supported by the National Socio-Environmental Synthesis Center (SESYNC).

## Additional information

### Funding

| Funder | Grant reference number | Author |
|---|---|---|
| Medical Research Council | G1100796 | Kimberly M Fornace<br>Neal Alexander<br>Tommy R Abidin<br>Paddy M Brock<br>Tock H Chua<br>Indra Vythilingam<br>Heather M Ferguson<br>Benny O Manin<br>Meng L Wong<br>Sui H Ng<br>Jon Cox<br>Chris Drakeley |

The funders had no role in study design, data collection and interpretation, or the decision to submit the work for publication.

### Author contributions

Kimberly M Fornace, Conceptualization, Data curation, Formal analysis, Investigation, Visualization, Methodology, Writing—original draft, Writing—review and editing; Neal Alexander, Validation, Methodology, Writing—review and editing; Tommy R Abidin, Data curation, Investigation; Paddy M Brock, Formal analysis, Methodology, Writing—review and editing; Tock H Chua, Data curation, Writing—review and editing, Interpretation of mosquito data; Indra Vythilingam, Data curation, Writing—review and editing, Interpretation and analysis of mosquito data; Heather M Ferguson, Methodology, Writing—review and editing; Benny O Manin, Data curation, Investigation, Writing—review and editing; Meng L Wong, Sui H Ng, Data curation, Investigation, Collection and analysis of mosquito data; Jon Cox, Conceptualization, Funding acquisition, Methodology, Writing—review and editing; Chris Drakeley, Conceptualization, Funding acquisition, Writing—review and editing

### Author ORCIDs

Kimberly M Fornace (iD) https://orcid.org/0000-0002-5484-241X
Tock H Chua (iD) http://orcid.org/0000-0002-8984-8723
Benny O Manin (iD) http://orcid.org/0000-0003-0726-6146
Chris Drakeley (iD) http://orcid.org/0000-0003-4863-075X

## Ethics

Human subjects: This study was approved by the Medical Research Sub-Committee of the Malaysian Ministry of Health (NMRR-12-537-12568) and the Research Ethics Committee of the London School of Hygiene and Tropical Medicine (6531). Written informed consent was obtained from all participants or parents or guardians and assent obtained from children under 18.

## Decision letter and Author response

Decision letter https://doi.org/10.7554/eLife.47602.026
Author response https://doi.org/10.7554/eLife.47602.027

## Additional files

### Supplementary files

• Source code 1. R scripts for fitting biased random bridges (with simulated GPS data), spatiotemporal models of mosquito biting rates and semi-continuous resource utilisation models.
DOI: https://doi.org/10.7554/eLife.47602.013

• Supplementary file 1. Data sources for assessed spatial and environmental covariates.
DOI: https://doi.org/10.7554/eLife.47602.014

• Supplementary file 2. Data sources of mosquito biting data.
DOI: https://doi.org/10.7554/eLife.47602.015

• Transparent reporting form DOI: https://doi.org/10.7554/eLife.47602.016

### Data availability

Data on human subjects is not available due to ethical restrictions around sharing identifiable information. All other data is publicly available with relevant links or publications included. Code to reproduce this analysis is available on GitHub or as supplementary information.

The following previously published datasets were used:

| Author(s) | Year | Dataset title | Dataset URL | Database and Identifier |
|---|---|---|---|---|
| NASA LP DAAC | 2015 | Landsat 8 Operational Land Imager | https://landsat.gsfc.nasa.gov/landsat-8/ | NASA Landsat Science, Landsat 8 |
| NASA LP DAAC | 2015 | Advanced Spaceborne Thermal Emission and Reflection Radiometer Global Digital Elevation Model | https://lpdaac.usgs.gov/products/astgtmv003/ | ASTER, 10.5067/ASTER/ASTGTM.003 |
| DAAC NL | 2015 | MODIS/ Terra Vegetation Indices 16-Day L3 Global 250m Grid SIN V006 | https://lpdaac.usgs.gov/products/mod13q1v006/ | MODIS, 10.5067/MODIS/MOD13Q1.006 |
| NASA TRMM | 2015 | Daily TRMM and other satellites precipitation product (3B42 V6 derived) | https://gcmd.nasa.gov/records/GCMD_GES_DISC_TRMM_3B42_daily_V6.html | Tropical Rainfall Measurement Mission, 10.5067/TRMM/TMPA/DAY/7 |

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
