## [Decision Letter]

Thank you for submitting your article "Local human movement patterns and land use impact exposure to zoonotic malaria in Malaysian Borneo" for consideration by *eLife*. Your article has been reviewed by three peer reviewers, including Ben Cooper as the Reviewing Editor and Reviewer #1, and the evaluation has been overseen by Neil Ferguson as the Senior Editor. The following individual involved in review of your submission has agreed to reveal their identity: Nick Ruktanonchai (Reviewer #3).

The reviewers have discussed the reviews with one another and the Reviewing Editor has drafted this decision to help you prepare a revised submission.

Summary:

This study builds on previous work establishing environmental risk factors for human infections with the zoonotic malaria *Plasmodium knowlesi* and uses human movement data obtained by GPS tracking, spatiotemporal models of mosquito distribution, and spatial land use data to provide a new perspective on *P. knowlesi* transmission. The analysis predicts that more than 90% of infectious bites will occur in areas surrounding households at forest edges.

Essential revisions:

The reviewers were agreed that this work potentially represents a significant advance to the *P. knowlesi* literature. However, there were concerns that some aspects of the work were not clearly motivated, and considerable attention to both the Materials and methods and Results is needed to ensure the work is clearly described, accessible to a broad audience and repeatable. Specifically, the following three points were considered essential revisions:

1) Provide a null model for comparison to illustrate added value of data.

Also, clarify the value of the BRB model over simpler approaches such as kernel smoothing or linear interpolation.

2) Fully describe the results about different land use patterns.

3) Clarify multiple aspects of methods and improve reporting.

A detailed list of specific points that need to be clarified can be found in the individual reviews below.

Reviewer #1:

This paper describes some interesting data concerning human movement patterns and there has been some sophisticated statistical modelling. However, I only have a superficial knowledge of the zoonotic malaria literature and, after carefully reading the manuscript, feel that I am not in a position to judge whether or not this work represents a substantial step forward in understanding the epidemiology of *P. knowlesi*.

1) I couldn't follow the model specification in the subsection “Human space use”:

e.g. these points were not clear

"we randomly sampled equal numbers of absences within the study site". I don't know what this means. Is this sampling from the data? What are the "equal numbers" equal to? These "absences" were sampled, but what was done with the samples? I may be missing something very obvious here, and perhaps this will be familiar to people working with similar models, but I could only begin to guess what is being done here and why.

Is "occurrence" the same thing as "presence" or something else? It hasn't been made clear what "occurrence" means here.

The first equation suggests "occurrence" for individual j in cell i is defined to be 1 if utilisation of that cell by the same individual is greater than 0. How does this relate (if at all) to the earlier statement that "absence" was defined as a grid cell with a UD of less than 0.00001?

I'm confused by the second equation as well, as the z_ij_ values are defined in terms of the y_ij_ values in the first equation, but the y_ij_ values are defined in terms of the z_ij_ values.

2) –Subsection “Exposure to infected vectors”, last paragraph. I couldn't make sense of this section. Specific questions:

i) r_it_, is a risk and so on the interval [0,1]. However, nothing in the first equation of this paragraph bounds the calculated r_it_ to this interval. Is the implicit assumption that r_it_ values will always be much less than 1 so that there is no need to worry about these bounds?

ii) How are a_jt_ and Z_j_ quantified, and how is uncertainty in these quantities accounted for?

Reviewer #2:

In 'Local human movement patterns and land use impact exposure to zoonotic malaria in Malaysian Borneo', the authors analyze a unique set of data to understand the role of human movement and land use patterns on the risk of exposure to *P. knowlesi* in Malaysia. *P. knowlesi* has unique ecological and epidemiological characteristics which makes the interdisciplinary approach (by capturing data about different environmental and human behavioral factors) particularly relevant. The authors use statistical models fit to GPS logger data from individuals, mosquito biting data, and geographic data to identify locations with the highest risk of *P. knowlesi* exposure. The primary result is that the majority of risk is nearby houses or forested areas, a useful finding that would be strengthened if more background about current public health interventions was provided. Given this result, there are points in the text where it appears that the manuscript was written for other results – based on these results it does not appear that the very detailed nature of daily movement patterns (which are quite variable) was not relevant to risk (since night time movement patterns were fairly consistent across groups). This is a useful and valuable finding, but often the text does not appear to emphasize the actual results (which should not be dismissed) presented.

Overall, there are three primary concerns about the current manuscript.

1) The authors do not provide a null model for comparison. Undoubtably, these data were time consuming and expensive to collect and it would not be feasible to collect these data systematically. However, the authors do not provide a reasonable null model as a point of comparison to clearly illustrate the added value of their data. For example, if you assumed that individuals spent 85%, 90%, 95%, etc. percentage of their time at their residence location instead of using the detailed GPS loggers, are the results substantially different? Similarly for the mosquito biting data, if you assumed that biting increased with forest cover, then how different are the results compared to the detailed analysis the authors provided? Since the final results are in terms of where there is the highest risk, it would be useful to see a comparison of where do these additional data and analyses add value by identifying locations of high risk that would have been missed otherwise. Without this type of comparison, which should be within the scope of the current article, it is difficult to assess how these results are different from a simpler model and set of assumptions. In general, it would be helpful if the authors highlighted where it would have been assumed that there was high risk if not for this analysis.

2) The authors do not fully describe their results about different land use patterns. Throughout the manuscript, the authors note that geographic variables about land use may not accurately reflect how individuals actually use the land. This is an important point and a useful topic to study for spatial analyses of infectious diseases that utilize these types of data. However, it is not clear exactly how their results translate to which types of patterns of use actually correspond to the geographic variables. For example, do certain geographic covariates correspond to different travel behavior? And in which instances do the different land use variables seem to better approximate similar travel behavior (within variable heterogeneity of use)?

3) Finally, it is unclear if the authors took into account or adjusted for their sampling scheme and the level of interpolation applied. Are there measures of uncertainty that could be applied to the analysis, perhaps informed by a null model? In addition, the total number of person nights appears to be highly correlated with the placement of the houses – which would suggest a spatial sampling bias, however there are a number of areas in red up to hundreds of kilometers from the houses. To what extent are these areas with a high number of person nights a function of the statistical smoothing or other covariates versus the raw data? It is understandable if the raw GPS coordinates are not made publicly available, but without additional information about the actual data and a more through presentation of these data, it is unclear to what extent these results are reflective of the actual travel patterns.

Reviewer #3:

Overall, this was a very well-written and interesting paper. The authors have done a generally good job writing up a pretty comprehensive set of analyses, and it represents an important step forward for modeling the spatial interaction of human movement patterns and vector borne disease transmission.

---

## [Author Response]

Essential revisions:The reviewers were agreed that this work potentially represents a significant advance to the P. knowlesi literature. However, there were concerns that some aspects of the work were not clearly motivated, and considerable attention to both the Materials and methods and Results is needed to ensure the work is clearly described, accessible to a broad audience and repeatable. Specifically, the following three points were considered essential revisions:1) Provide a null model for comparison to illustrate added value of data.

We can appreciate this request and have included further details of a model of mosquito biting and infection rates not incorporating human movement to address this concern in the reviewer specific comments below (response to reviewer 2, point 1).

Also, clarify the value of the BRB model over simpler approaches such as kernel smoothing or linear interpolation.

We have included substantially more background on the BRB model in the Introduction (Introduction, sixth paragraph), including advantages compared with kernel density smoothing. Specifically, this includes using the GPS points as a time-ordered series of points to more accurately estimate space use when there are missing data or irregular time intervals.

2) Fully describe the results about different land use patterns.

We have clarified the difference between land use and land cover and fully described the aspect of land use we have focused on (i.e. time spent in different land cover types) (Introduction, second and last paragraphs). The full results of this analysis are included in the predictive movement model (Table 2) and we have described additional land use topics which could be addressed in the Discussion (e.g. behaviours or activities in different locations) in the Discussion (last paragraph). We have additionally included a map of the study sites and land cover (Figure 2). Full details on classification of land cover are included in Fornace et. Al., 2018.

3) Clarify multiple aspects of methods and improve reporting.

We have substantially revised the methods and reporting throughout the manuscript. Specific points have been included below.

Reviewer #1:1) I couldn't follow the model specification in the subsection “Human space use”:

We thank the reviewer for these comments and we have substantially revised the model description to clarify this (Materials and methods). We appreciate the time taken to review these methods in detail.

e.g. these points were not clear"we randomly sampled equal numbers of absences within the study site". I don't know what this means. Is this sampling from the data? What are the "equal numbers" equal to? These "absences" were sampled, but what was done with the samples? I may be missing something very obvious here, and perhaps this will be familiar to people working with similar models, but I could only begin to guess what is being done here and why.

We have reworded this section to clarify these issues in the last paragraph of the subsection “Human space use”. This model includes two components: 1) a binomial model of whether or not the grid cell was visited, with presence defined as UD > 0.00001 and absence defined as UD < 0.00001; and 2) a γ model of the proportion of time spent at a grid cell if it was visited by an individual.

To fit the binomial component of this model, we needed to define presences and absences (i.e. places visited and not visited). As the UD is a probability distribution, it does not reach 0 and we considered all grid cells having a UD of less than 0.00001 as absences, i.e. we considered this value as negligible. To fit a binomial model of grid cells an individual visited, we included all grid cells visited (UD > 0.00001) and randomly subsampled an equal number of grid cells not visited. Within species distribution models, equal numbers of absences (or pseudoabsences) are commonly sampled to improve model predictive ability and we have included a reference describing this. This is described in the last paragraph of the subsection “Human space use”.

Is "occurrence" the same thing as "presence" or something else? It hasn't been made clear what "occurrence" means here.

We apologise that this was unclear and we have reworded this to be “presence” throughout. Specifically, this refers to the probability of grid cells being visited by an individual during the sampling period.

The first equation suggests "occurrence" for individual j in cell i is defined to be 1 if utilisation of that cell by the same individual is greater than 0. How does this relate (if at all) to the earlier statement that "absence" was defined as a grid cell with a UD of less than 0.00001?

Please see our second response to point #1 above.

I'm confused by the second equation as well, as the z_ij_ values are defined in terms of the y_ij_ values in the first equation, but the y_ij_ values are defined in terms of the z_ij_ values.

We have edited the symbols used for clarity and have revised the hurdle method description to clarify the utilisation distribution probability of a grid cell is conditional of an individual having visited this grid cell (subsection “Human space use”, last paragraph). We have also edited all equation symbols used for consistency (Materials and methods).

2) Subsection “Exposure to infected vectors”, last paragraph. I couldn't make sense of this section. Specific questions:i) r_it_, is a risk and so on the interval [0,1]. However, nothing in the first equation of this paragraph bounds the calculated r_it_ to this interval. Is the implicit assumption that r_it_ values will always be much less than 1 so that there is no need to worry about these bounds?

We have revised the wording around this to clarify that this refers to an exposure risk, specifically referring to the number of infectious bites received by an individual or the total of infectious bites occurring within a particular location (grid cell). As such, this quantity could theoretically range from 0 to infinity (subsection “Exposure to infected vectors”, last paragraph).

ii) How are a_jt_ and Z_j_ quantified, and how is uncertainty in these quantities accounted for?

These were calculated as derived quantities from the 3 components: human space use (*y*), mosquito biting rates (*m*) and infection rate (*c*) (Figure 1). The notation used around this was not clear in the earlier draft so we have edited to ensure consistency. In addition, we have included the details of the mosquito sporozoite data used to estimate proportion of infectious mosquitoes (–subsections “Human space use” and “Exposure to infected vectors”). As these numbers were very low, it was not possible to model variation in infection rates across time and space and we have included this as a limitation in the Discussion (Discussion, last paragraph).

Reviewer #2:In 'Local human movement patterns and land use impact exposure to zoonotic malaria in Malaysian Borneo', the authors analyze a unique set of data to understand the role of human movement and land use patterns on the risk of exposure to P. knowlesi in Malaysia. […] This is a useful and valuable finding, but often the text does not appear to emphasize the actual results (which should not be dismissed) presented.

We can appreciate the reviewer’s point and have included additional results to clarify. Specifically, while most of the night time movements are predictable (results described in Table 2 and Results), there were still substantial variations in the predicted numbers of infectious bites (Table 5 and Results).

Overall, there are three primary concerns about the current manuscript.1) The authors do not provide a null model for comparison. Undoubtably, these data were time consuming and expensive to collect and it would not be feasible to collect these data systematically. However, the authors do not provide a reasonable null model as a point of comparison to clearly illustrate the added value of their data. For example, if you assumed that individuals spent 85%, 90%, 95%, etc. percentage of their time at their residence location instead of using the detailed GPS loggers, are the results substantially different? Similarly for the mosquito biting data, if you assumed that biting increased with forest cover, then how different are the results compared to the detailed analysis the authors provided? Since the final results are in terms of where there is the highest risk, it would be useful to see a comparison of where do these additional data and analyses add value by identifying locations of high risk that would have been missed otherwise. Without this type of comparison, which should be within the scope of the current article, it is difficult to assess how these results are different from a simpler model and set of assumptions. In general, it would be helpful if the authors highlighted where it would have been assumed that there was high risk if not for this analysis.

We can appreciate the reviewer’s point about the need for comparison with a null model. In choosing such a model, we do not feel it would be valid to compare with estimates of risk assuming individuals spend X% of time around the house; while these might be reasonable assumptions to make based on this data, these assumptions cannot be made without collecting this type of data and the purpose of this study was to examine the distribution of these movements precisely because these data do not exist in the context of zoonotic malaria. We have included a sentence in the Discussion highlighting the need to consider measures human space when defining risk areas (third paragraph).

Rather, to highlight the importance of including movement, we have included additional data about where infectious mosquito bites would be expected based only on mosquito distribution data, serving as a null model (Results, last paragraph). This is additionally illustrated in Figure 5, where areas with higher mosquito biting rates (5C) do not correspond to areas with higher probabilities of human use (5D). This model highlights the difference in areas which would be identified using only standard vector surveillance vs. areas identified by also including human movement. Human landing catch data on mosquito biting rates are a core component of surveillance programmes and used to guide interventions. By also considering human space use, this highlights importance of areas with lower mosquito biting rates but much higher probabilities of human presence.

2) The authors do not fully describe their results about different land use patterns. Throughout the manuscript, the authors note that geographic variables about land use may not accurately reflect how individuals actually use the land. This is an important point and a useful topic to study for spatial analyses of infectious diseases that utilize these types of data. However, it is not clear exactly how their results translate to which types of patterns of use actually correspond to the geographic variables. For example, do certain geographic covariates correspond to different travel behavior? And in which instances do the different land use variables seem to better approximate similar travel behavior (within variable heterogeneity of use)?

Land use is commonly defined as the “total of arrangements, activities and inputs that people undertake in a certain land cover type” (IPCC, 2000). Within this broad definition, we have specifically focused on one aspect of land use, human use and movement into different environments (Introduction, last paragraph). This is a core component of this analysis and led to the development of a predictive model including the probability individuals would visit a specific location and, if the location was visited, the proportion of time spent at this location (Materials and methods, subsection “Human space use”). To develop this model, we specifically evaluated a number of geographic covariates, including distance to own house, distance to closest house, distance to roads, land cover class, distance to forest edge, elevation and slope (subsection “Human space use”, last paragraph). The full results of the final model and influence on space use are included in Table 2.

This model specifically describes one aspect of travel behaviour, the probability of visiting a location and the time spent in this location. If, by travel behaviour, this reviewer is referring to the activities undertaken by individual in a specific location, this is beyond the scope of this study as GPS data is specifically used to quantitatively measure the locations visited and the time spent in these locations. Behaviours in different environments may be better assessed using qualitative methods or detailed travel diaries on activities undertaken within different areas.

To clarify any confusion between land cover (the physical terrestrial surface) and land use (human activities undertaken on these surfaces), we have edited the Introduction and included the IPCC reference on the definition. We have also clarified that we are specifically focusing on one specific aspect of land use which is most relevant for infection dynamics (Introduction, second and last paragraphs). We have additionally added a sentence in the Discussion about the potential for future behavioural studies to explore other aspects of land use (Discussion, fourth paragraph) and have edited the wording to further clarify.

3) Finally, it is unclear if the authors took into account or adjusted for their sampling scheme and the level of interpolation applied. Are there measures of uncertainty that could be applied to the analysis, perhaps informed by a null model?

Yes, we did adjust for the sampling scheme. Although a total of 285 randomly selected individuals were sampled (roughly 20% of the populations over the age of 8 within these sites), predictive models used data for all individuals residing within these communities. We have substantially revised the Materials and methods section to clarify these points, particularly in relation to the characteristics of the community and individuals sampled for GPS tracking (Table 1, Figure 2). The level and method of interpolation used to estimate utilisation distributions are described in the sixth paragraph of the Introduction and Materials and methods subsection “Human space use”), with source code included as Source code 1. This includes the parameters used for smoothing and interpolating missing data.

In addition, the total number of person nights appears to be highly correlated with the placement of the houses – which would suggest a spatial sampling bias, however there are a number of areas in red up to hundreds of kilometers from the houses. To what extent are these areas with a high number of person nights a function of the statistical smoothing or other covariates versus the raw data? It is understandable if the raw GPS coordinates are not made publicly available, but without additional information about the actual data and a more through presentation of these data, it is unclear to what extent these results are reflective of the actual travel patterns.

We have included additional details on the demographic characteristics of the individuals included within the GPS tracking study and the study site population used for predictive models (Table 1). To address concerns about spatial bias, we have additionally included a map of the two study sites, with locations of households included in the GPS tracking study and survey (Figure 2). Further details on the individual household locations and GPS points could not be included due to ethics restrictions.

The prediction of high numbers of person nights spent in close proximity to houses is not an artefact of sampling bias but rather because of the effects of the covariates. One of the main predictors of human movement between 6pm and 6am is Euclidean distance from the individuals own house, followed by distance from other houses and roads. The predictive model was limited to areas within 5km of the households included in the study site (subsection “Human space use”, last paragraph) for all individuals aged 8 and over residing in this community. We have included an additional line in the Discussion that this does not allow assessment of long range or infrequent movements (third paragraph).

Regarding concerns about spatial smoothing, we have included further detail on using biased random bridges to estimate utilisation distributions (Introduction, sixth paragraph) and the specific parameters used (subsection “Human space use”), with an example script to run this analysis included in Source code 1. We used a minimum smoothing parameter of 30m to account for the resolution of habitat data (land cover and other predictors). The effects of these parameters have also been discussed in referenced literature (Benhamou, 2011).